# COVID-19 Literature Topic-Based Search via Hierarchical NMF

**Rachel Grotheer**[1]**, Longxiu Huang**[3]**, Yihuan Huang**[3]**, Alona Kryshchenko**[2]**, Oleksandr Kryshchenko**[4]**,
Pengyu Li**[3]**, Xia Li**[3]**, Elizaveta Rebrova**[3]**, Kyung Ha**[3]**, Deanna Needell**[3]

[1] Department of Mathematics, Wofford College, Spartanburg, SC
[2] Department of Mathematics, California State University Channel Islands, Camarillo, CA
[3] Department of Mathematics, University of California, Los Angeles, CA
[4] LWS Research, West Hollywood, CA
grotheerre@wofford.edu,
{kyungha,charlotte0408,erby1215}@g.ucla.edu, {huangl3,xli5,deanna,rebrova}@math.ucla.edu,
alona.kryshchenko@csuci.edu, okryshchenko@lwsresearch.com.

## Abstract

A dataset of COVID-19-related scientific literature is compiled, combining the articles from several online libraries and selecting those with open access and full text available. Then, hierarchical nonnegative matrix factorization is used to organize literature related to the novel coronavirus into a tree structure that allows researchers to search for relevant literature based on detected topics. We discover eight major latent topics and 52 granular subtopics in the body of literature, related to vaccines, genetic structure and modeling of the disease and patient studies, as well as related diseases and virology. In order that our tool may help current researchers, an interactive website is created that organizes available literature using this hierarchical structure.

## 1 Introduction

The appearance of the novel SARS-CoV-2 virus on the global scale has generated demand for rapid research into the virus and the disease it causes, COVID-19. However, the literature about coronaviruses such as SARS-CoV-2 is vast and difficult to sift through. This paper describes an attempt to organize existing literature on coronaviruses, other pandemics, and early research on the current COVID-19 outbreak in response to the call to action issued by the White House Office of Science and Technology policy (Science and Policy, 2020) and posted on the Semantic Scholar (Scholar, 2020)

and Kaggle (2020) websites. The original dataset posted on that site is augmented by adding articles drawn from other databases in order to make the final interactive organizational structure more robust for researchers.

Our primary goal is to create a framework for a topic-based search of papers within this dataset that is helpful to those investigating the novel coronavirus, SARS-CoV-2, and the global COVID-19 pandemic. In order to discover the latent topics present in the collection of scholarly articles, as well as to organize them into a hierarchical tree structure that allows for an interactive search, we use a modified hierarchical nonnegative matrix factorization (HNMF) approach. A website[1] that allows users to walk through the topic tree based on the top keywords associated with each topic is created using this hierarchical organization of the papers.

### 1.1 Contributions

Our methods help make sense of a vast and rapidly growing body of COVID-19 related literature. The main contributions of this paper are as follows:

- A diverse dataset of COVID-19 related scientific literature is compiled, consisting of articles with full-text available drawn from several online collections.

---

[1] http://covid-19-literature-clustering.net/

- A tree-like soft[2] cluster structure is created of all the papers in the dataset based on the inherent relation between their topics using hierarchical NMF.

- The best number of topics for each layer is defined as the number that produces the most consistent clustering of the dataset with random initializations of NMF algorithm. A variance analysis method is used to identify the best number of topics on each layer.

- The effectiveness of the method is measured by exploring the coherence of each topic and dissimilarity between the topics.

- The discovered topics and distribution of articles into each of the topics are discussed, revealing the major areas of interest and research in the early months of the pandemic, as well as how existing epidemic literature can be effectively organized to allow efficient comparison to COVID-19 related research.

- The theoretical results are complemented with an interactive website.

## 1.2 Related Work

Some relevant works that motivate our approach are briefly reviewed. NMF was first proposed for document clustering (Xu et al., 2003), and since then many variants of the NMF algorithm have been proposed and applied to help organize various types of data (Lee and Seung, 1999; Buciu, 2008; Kuang et al., 2015). In particular, there exist several recent papers that use NMF to find a hierarchy of topics in a set of documents. For example, Kuang and Park (2013) apply a rank-2 NMF to the recursive splitting of a text corpus and also provide an efficient on-the-fly stopping criterion. Gao et al. (2019) discuss a different version of HNMF, when the hierarchy of topics is generated by aggregation of the topics (rather than splitting). The first application of NMF produces the initial set of the most refined topics, and the subsequent NMF iterations find supertopics in which the previous set of topics can be summarized. This approach is referred as a *bottom-to-top* viewpoint, and the former as a *top-to-bottom*. Approaches that utilize tools from neural networks such as back propagation to improve the

topic representations have also been developed recently (Trigeorgis et al., 2016; Le Roux et al., 2015; Sun et al., 2017; Gao et al., 2019).

Tu et al. (2018) propose a hierarchical online non-negative matrix factorization method (HONMF) to generate topic hierarchies from data streams. The proposed method can dynamically adjust the topic hierarchy to adapt to the emerging, evolving and fading process of the topics. This work most closely aligns with what we present here, and although we do not consider the online setting, our method can easily be adapted to such.

Finally, several authors have sought to address the issue of interpretability of topics discovered by NMF, especially in datasets comprised of text documents. For example, Ailem et al. (2017) apply NMF to the documents using a word embedding model, *Word2Vec* (Mikolov et al., 2013b), that focuses on the semantic relationship between words. We make use of this embedding to analyze the usefulness of the topics generated by examining their semantic similarity.

## 2 Data Description

The dataset used is compiled from 4 different databases that contain scholarly articles related to COVID-19, various coronavirus diseases, other infectious diseases, and epidemiology (Scholar, 2020; for Disease Control and Prevention, 2020; National Center for Biotechnology Information, 2020; bioRxiv, 2020).

From each of these databases, only articles written in English that have a complete abstract and text body available are included. Punctuation and words on the NLTK English stopwords list (Bird et al., 2009) are removed from the text body and abstract of each article. An initial application of NMF generated a topic consisting primarily of words that, upon further investigation, were found to be part of the copyright and publishing information present at the top of articles primarily drawn from the bioRxiv database, and not the content of the articles themselves. Therefore, we also remove the top 30 keywords of that topic from the corpus. See Appendix 7.1 for the list of these removed keywords. Finally, the articles are lemmatized and each word in the text body and abstract is represented by a TD-IDF embedding (Salton and Buckley, 1988). After processing and cleaning, the final dataset contains 25,663 articles. Most of these databases are regularly updated and one of the important future

---

[2]*soft* here means that clusters can intersect, as one paper could belong to more than one topic

directions of this work will include developing a dynamic tree structure that pulls new articles from these databases weekly.

# 3 Hierarchical NMF for Topic Detection

In a vector space model, a corpus can be represented by a $d \times n$ matrix $X$, where $d$ is the size of the vocabulary, and $n$ is the number of documents. The underlying assumptions in topic modeling (Blei et al., 2007) are that a latent topic can be represented as a distribution over the words, and that every document is a mixture of topics, i.e. comprises a statistical distribution of topics that can be obtained by "adding up" all of the distributions of all the topics covered. In this section, we will introduce how to apply hierarchical NMF for topic detection and creation of the hierarchical tree structure. As a preliminary step, a brief introduction to using NMF for topic detection is given.

## 3.1 NMF for Topic Detection

In NMF, the corpus matrix $X \in \mathbb{R}_{\geq 0}^{d \times n}$ is decomposed into a pair of low-rank nonnegative matrices $W \in \mathbb{R}^{d \times k}$, also known as the dictionary matrix, and $H \in \mathbb{R}^{k \times n}$, also known as the coding matrix, by solving the following optimization problem

$$\inf_{W \in \mathbb{R}_{\geq 0}^{d \times k}, H \in \mathbb{R}_{\geq 0}^{k \times n}} \|X - WH\|_F^2, \qquad (1)$$

where $\|A\|_F^2 = \sum_{i,j} A_{ij}^2$ denotes the matrix Frobenius norm.

NMF, essentially an iterative optimization algorithm, has a drawback: the objective function is usually non-convex and has multiple local minima. Therefore a different random initialization of the NMF procedure will result in a different matrix factorization. More importantly, this changes the interpretation of the results, including topic vector representations ($W$) as well as the relevance between articles and topics ($H$). Another possible source of variability in the algorithm is the choice of the number of topics, $k$. Different combinations of initializations of $W$, $H$, and $k$ yield different topics, leading to different article clustering results. See Section 5.1 for more discussion and implementation details in this vein.

## 3.2 Hierarchical NMF

The traditional NMF method treats the detected topics as a flat structure, which limits the ability of the representation of such method. In contrast, a hierarchical NMF (HNMF) framework is able to detect supertopics, subtopics, and the relationship between them, creating a tree structure. Compared with traditional NMF, HNMF improves topic interpretability. For instance, while both NMF and HNMF may produce topics that are similar to one another, if these topics are in the bottom layer of the tree structure provided by HNMF, their associated supertopics provide additional context to help distinguish the related subtopics. Besides improving topic interpretability, HMNF also provides a more user-friendly search framework, which is suitable for building website. The hierarchy allows users to search for relevant topics more effectively, while progressively narrowing their search.

Given the complex nature of the coronavirus literature corpus, such a hierarchical approach is appealing. Thus, we apply the HNMF algorithm summarized in Algorithm 1. Note that this algorithm is similar to the one in (Tu et al., 2018), which has been shown to be effective for topic detection.

In HNMF, NMF is first applied to the original corpus matrix $X$ to obtain the dictionary matrix $W$ and coding matrix $H$. The documents are then sorted into matrices $X_1, X_2, \cdots, X_k$, each representing a different topic, according to the coding matrix $H$, or into the matrix $X_e$ that temporarily holds unassigned articles. Whether the leaves need to be further divided depends on the number of the documents in each topic matrix (leaf). If the number of documents sorted into a topic is greater than a pre-specified value $m$, then a further division is needed. The above process is repeated until the number of documents in each leaf is less than $m$. More details on the implementation of the HNMF algorithm are provided in Section 5.2.

# 4 Discussion of Results

This section begins with a discussion and visualization of the hierarchical tree structure obtained using Algorithm 1. Then in Sections 4.3 and 4.4 quantitative evidence is provided that the discovered topics are reasonable. In doing this, we seek to measure both the rationality of a given topic and the similarity between topics to evaluate whether the topics differ enough to be useful for a user.

## 4.1 Topic Visualization

Implementation of Algorithm 1 on the dataset results in a hierarchical clustering of the articles into eight supertopics, each with five to six

**Algorithm 1:** Hierarchical NMF

**Input:** Corpus matrix $X$.

$[W, H] = \text{NMF}(X, k^*)$ where topic number $k^*$ is chosen by Algorithm 2;

assign articles to the related topics $X_1, \cdots, X_{k^*}$ according to the threshold $\alpha$ in $H$, and any remaining articles to "Extra Document" matrix $X_e$;

**while** *# of the articles assigned to a topic* $i > m$ **do**

    determine the # of sub-topics $k_i^*$ of the topic $i$ in $X_i$ by Algorithm 2;

    $[W_i, H_i] = \text{NMF}(X_i, k_i^*)$;

    assign the documents to the topics by the a threshold $\alpha$ in $H_i$s;

    assign the rest to $X_e$;

**end**

**for** *article $x_i$ in $X_e$* **do**

    calculate cosine similarity between $x_i$ and leaves, and assign the article to the most related leaf;

**end**

repeat both while and for loops until the number of the articles assigned to each topic is less than m.

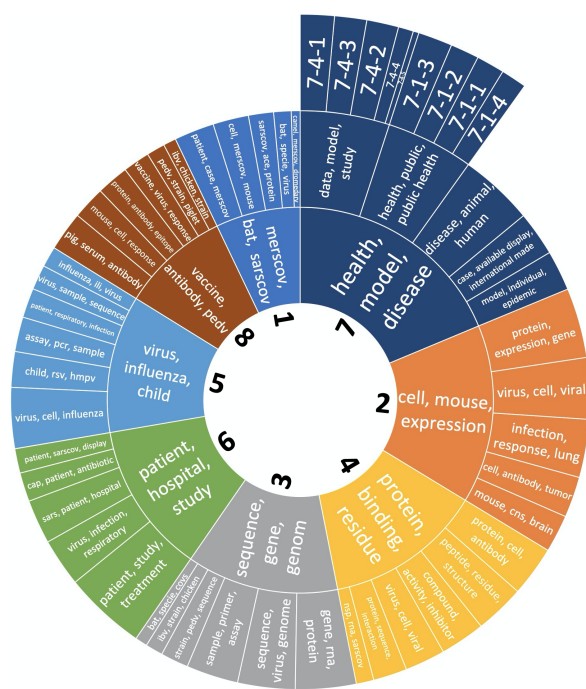

Figure 1: Sunburst Diagram of the complete hierarchical structure. The top three relevant words per topic are shown. The area of each region is proportional to the number of articles in that topic. See appendix for the keywords associated with the third layer. The inner circle numeric labels are corresponding to topic number in Figure 2

subtopics. Two of these subtopics, the first and fourth subtopics of supertopic 7, are further decomposed into a third layer of subtopics as the number of articles assigned to the first and fourth subtopics are larger than the selected $m$ in Algorithm 1. The full hierarchical tree structure is visualized in the diagram in Figure 1. Each color represents one of the eight supertopics and the size of each slice is proportional to the number of articles that are clustered into that topic. It is important to note that only the top three words associated to each topic are shown due to space constraints, but in some cases extending the list of highly related words is necessary to clarify the difference between the subtopics. For reference, the top ten keywords associated with each topic and subtopic can be found in Appendix 7. Additionally, the five most probable words associated to each topic are displayed on the associated website to aid users in more effectively choosing the topics of personal interest.

In order to examine the structure in more depth, Figure 2 displays a branch of the resulting tree represented by word clouds, generated from the top five words associated with each topic. The size of the words in each word cloud cell are proportional

to their weight in the corresponding $W$ matrices, and thus, the probability they are associated with that topic. In particular, the figure follows one path down the tree structure, focusing on Topic 7 and its associated subtopics, and then continuing to the subtopics of Topic 7-1. When moving to deeper layers in the tree, the general "health" and "model" topic further differentiates into subtopics ranging from public health to animal to human transmission diseases, and data modeling. Finally, the public health subtopic leads to clusters of articles specifically related to China or hospital care, for example.

## 4.2 Discussion of Topics

Perhaps not surprisingly, the topic to which the highest number of articles are assigned, Topic 7, is about the general study of the disease (with the most highly associated words being "health, model, disease, case, epidemic, outbreak, public, country, population, transmission"), further split into two additional layers of subtopics. This is the only topic that was split into a third layer, allowing a more effective differentiation between articles covering

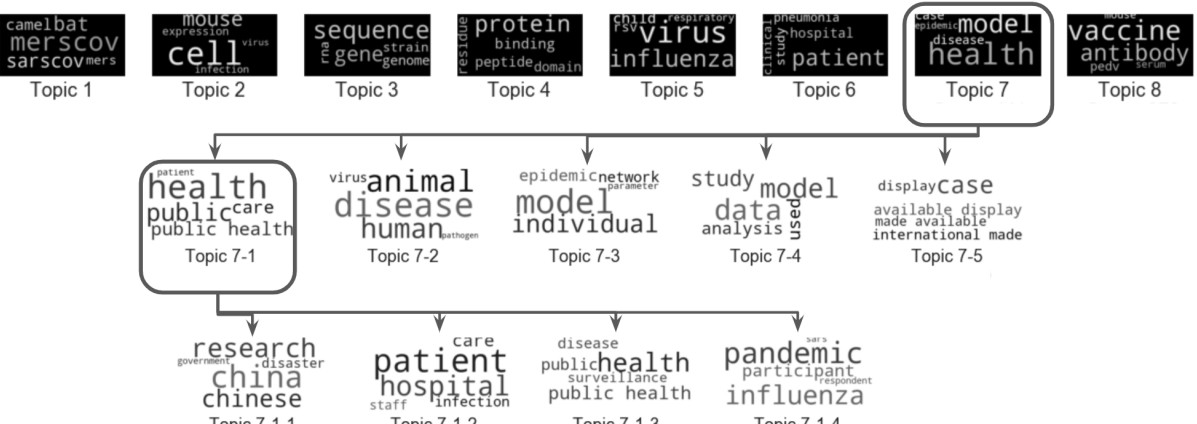

Figure 2: Part of Topics from HNMF and related topic coherence: The first row shows the the key words for the topics in the first layer, the second row shows the subtopics of Topic 7 and the subtopics of Topic 7-1 is showed in row 3. Corresponding *topic coherence* score (see Section 4.3 for more details) is underneath each word cloud.

a similar topic.

Also unsurprisingly, much of the literature, which was compiled early on during the pandemic, is clustered around the study of other coronavirus-caused diseases. Topic 8, for example, focuses on vaccine development through the lens of the Porcine Epidemic Diarrhea Virus (PEDV). Although this is a coronavirus found only in pigs, several vaccines have been developed, especially within the last seven years, when PEDV was first discovered in North America (Gerdts and Zakhartchouk, 2017). Hence, it is reasonable that this topic would be of interest to current researchers looking to develop a vaccine for SARS-CoV-2. Similarly, Topic 1 focuses on coronaviruses known to infect humans, such as SARS-CoV, and MERS-CoV. Topic 4 also contains a couple of subtopics that look specifically at the genetic structure of SARS-CoV.

Other topics of interest focus on articles about diseases with related symptoms, although they may be caused by a different type of virus. For example, both Topics 5 and 6 examine literature related to respiratory illnesses such as influenza, though Topic 5 clusters articles more related to laboratory study and Topic 6 clusters articles more related to hospital studies and patient care.

Other major topics focus more on microbiology, including the genomic structure of the virus, the cellular infection and immuno-response, and cell-protein interaction. Thus, the hierarchical tree structure separates papers between macro- (public health) and micro- (biological) studies of the virus, and into papers that study related viruses. This creates a clear delineation of topics for those investigating papers, and gives insight into areas of interest for early researchers of SARS-CoV-2. This organizational structure appears to be more robust and high-level than e.g. a keyword based search or organization.

## 4.3 Topic Coherence

One measure of effectiveness of the topics discovered by HNMF is *topic coherence*. Topic coherence is a quantitative measure of how well the keywords that define a topic make sense as a whole to a human observer and collectively provide a consistent interpretation of the topic.

While many topic coherence measures have been proposed, Röder et al. (2015) found that the $C_V$ coherence metric correlates the most closely with evaluation by human experts. The $C_V$ measure calculates the similarity between two words $w_i$ and $w_j$ using the normalized pointwise mutual information (NMPI) metric defined as,

$$
\text{NPMI}(w_i, w_j)^\gamma = \left( \frac{\log \frac{\mathcal{P}(w_i, w_j) + \epsilon}{\mathcal{P}(w_i) \cdot \mathcal{P}(w_j)}}{-\log(\mathcal{P}(w_i, w_j) + \epsilon)} \right)^\gamma
$$
(2)

where $\mathcal{P}(w_i)$ and $\mathcal{P}(w_i, w_j)$ are probabilities defined as the number of documents in which either $w_i$ or $(w_i, w_j)$, respectively, appear, divided by the total number of documents. These probabilities are calculated using a sliding Boolean window of size $s$ that slides over a document at the rate of one word per step. The sliding window allows for the proximity of the words to be taken into account. The $\gamma$ allows for more weight to be placed on higher

| Topic | $C_V$ | C |
|:-----:|:-----:|:---:|
| 1 | 0.68 | 321 |
| 2 | 0.63 | 442 |
| 3 | 0.68 | 407 |
| 4 | 0.56 | 419 |
| 5 | 0.61 | 405 |
| 6 | 0.66 | 420 |
| 7 | 0.63 | 441 |
| 8 | 0.59 | 378 |

Table 1: The coherence scores based on both the $C$ and $C_V$ metric for each of the 8 topics in the first layer of the tree.

NPMI values. After the NPMI score is calculated between each of the top $N$ words, $W'$, in each topic $W = \{W_1, W_2, \ldots, W_N\}$ and each of the remaining $N - 1$ words, $W^*$, these scores are added together to form a context vector $\vec{v}(W')$. Using the notation given by Syed and Spruit (2017),who applied the $C_V$ metric to topics found using latent Dirichlet allocation (LDA), we define the context vector as,

$$\vec{v}(W') = \left\{ \sum_{w_i \in W'} \text{NPMI}(w_i, w_j)^{\gamma} \right\}_{j=1,\ldots,N} \quad (3)$$

Finally, the cosine similarity between all context vector pairs within $S_i = (W', W^*)$ is calculated, giving the confirmation measure $\phi_{S_i}$,

$$\phi_{S_i}(\vec{u}, \vec{w}) = \frac{\sum_{i=1}^{N} u_i \cdot w_i}{||\vec{u}||_2 \cdot ||\vec{w}||_2} \quad (4)$$

which is a measure of how well word $W'$ in topic $W$ is supported by the word $W^*$ relative to all the words in $W$.

To further support these results, we additionally calculate the coherence score defined by Mimno et al. (2011) for each topic. The coherence score $C_i$ for topic $i$, $i = 1, \ldots, k$ is given by,

$$C_i(W^{(i)}) = \sum_{p=2}^{N} \sum_{\ell=1}^{p-1} \log \frac{P(w_p^{(i)}, w_\ell^{(i)}) + 1}{P(w_\ell^{(i)})}. \quad (5)$$

The topic coherence scores for each of the topics in the first layer, using both the $C_V$ and $C$ metrics are in Table 1.

The $C_V$ coherence metric has values between 0 and 1, with values closer to one indicating that the keywords form a topic that would be highly ranked by human expert. A positive, large coherence score using the $C$ metric indicates the same. A coherence score that is close to 0 (for $C_V$) or negative (for $C$) indicates that a topic is less meaningful, which may occur, for example, if the associated keywords fall into two unrelated groups, or if the keywords are seemingly random and have no obvious connection. Most of our identified subtopics have coherence scores whose values suggest that they are understandable and useful to human users. The $C_V$ scores for each of the subtopics can be found in Appendix 7.2.

### 4.4 Topic Similarity

Another test of the usefulness of the hierarchical structure generated is to evaluate whether the topics are different enough to allow for informative choice between them. To evaluate this, we quantify topic similarity using a metric known as the Word Mover's Distance (WMD). WMD is a popular tool for measuring distances between documents (Kusner et al., 2015). WMD utilizes *Word2Vec* (Mikolov et al., 2013a), a word embedding technique, and treats each document as a set of vectors in the embedded vector space. This embedding allows the WMD metric to consider the semantic meaning of a given word, rather than just its spelling. Thus, for example, it allows for identification of synonyms as having the same meaning in a given context despite being different words , which makes it more preferable than traditional metrics such as cosine similarity or Euclidean distance. The distance between two documents A and B is defined as the minimum cumulative distance that words from document A need to travel to match exactly the words of document B. We note that while there are other state of the art semantic representations, such as BERT (Devlin et al., 2018) and ELMo (Peters et al., 2018), and associated metrics, since the topics extracted are a bag of words with weights, the WMD with Word2Vec is sufficient for our purposes.

The topic similarity across the layers and within each layer is evaluated by computing the WMD between a topic and its associated subtopics and between the subtopics themselves, where each topic is represented by its 100 most related words. The similarities between all topics in the hierarchical structure obtained from HNMF is visualized in the heat map in Figure 3. As indicated by the overall dark colors, in general each topic in the tree is dissimilar from the others.

When examining the similarities between a topic and its subtopics, results show that for a given topic, its subtopics are less correlated with each other than with their parent topic. For example, in Figure 4, for Topic 7, the similarity scores between its subtopics are much lower than the scores between subtopics and their parent Topic 7. Similar results can be drawn for Topic 7-1 and its subtopics, as shown in Figure 5.

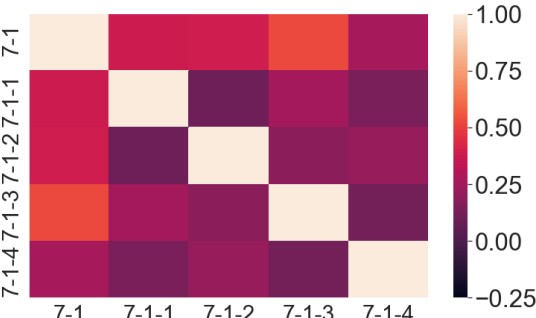

Figure 5: Topic similarity between Topic 7-1 and its subtopics measured by WDM: Topic 7-1 has high topic similarity with its four subtopics (7-1-1, 7-1-2, 7-1-3, 7-1-4) and the four topics have low similarity between themselves.

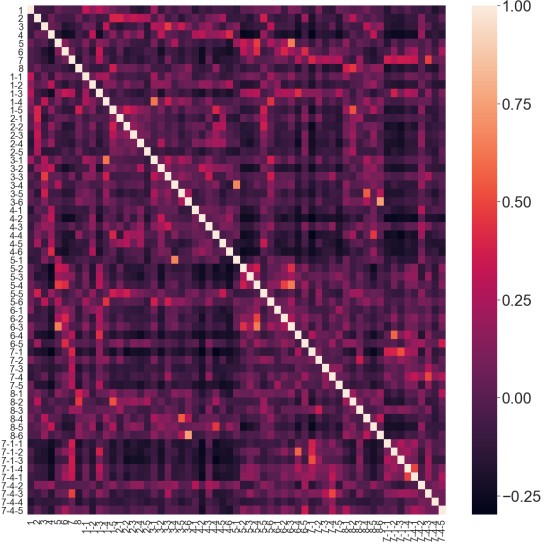

Figure 3: Topic similarity for all the topics from HNMF measured by WDM . A dark color indicates the topics are dissimilar, while a light color indicates high similarity. Note that the topics are listed from first layer to third layer from top to bottom or right to left on the vertical and horizontal axes, respectively.

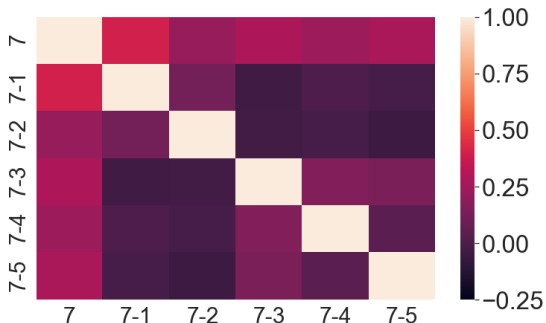

Figure 4: Topic similarity between Topic 7 and its subtopics measured by WDM: Topic 7 has high topic similarity with its five subtopics (7-1, 7-2, 7-3, 7-4, 7-5) and the five topics have low similarity between themselves.

However, there are some high similarity scores between subtopics that belong to different topics, for example the light off-diagonal spot in Figure 3 showing the similarities between Topics 6-3 and 5-3. Examining the top ten keywords associated with

each topic, we find that both topics are associated with the words "influenza", "virus", and "study" indicating that both topics deal with studies related to the influenza virus.

The insight into the difference between the two subtopics comes from examining supertopics 5 and 6 and the keywords associated with each subtopic that do not overlap. Looking at words such as "detection" and "assay" associated with Topic 5 and "surveillance", "case", "season", and "year" associated with Topic 5-3, it appears that Topic 5-3 is more associated with detecting and monitoring the prevalence of cases of influenza in the general populace in a given flu season. On the other hand, the presence of keywords "patient", "hospital", "clinical", and "study" associated with the parent topic, Topic 6, as well as "patient", "child", and "respiratory" associated with Topic 6-3, it seems that Topic 6-3, while also related to influenza studies, deals more specifically with cases in a hospital setting, perhaps specifically related to children, and examining the relationship with respiratory illness in general.

A study of similar subtopics such as these show the effectiveness of the tree in separating related topics into more dissimilar supertopics to make navigation to articles of interest clear. However, Algorithm 1 allows for an article to be assigned to more than one subtopic, acknowledging that a single article may of equal interest to researchers investigating different, but related topics.

## 5 Implementation

In this section, we discuss the details of the implementation of HNMF and the construction of the

**Algorithm 2:** Determine optimal number of topics

**Input:** integer $q$, corpus matrix $X$.

Determine a range for the potential topic number $[k_1, k_2]$ by plotting increment in variance explained by adding one more cluster to $X$;

randomly select $q + 1$ seeds for initialization;

**for** *integer $k$ in $[k_1, k_2]$* **do**

    generate topic sets $\{T_j\}_{j=1}^{q+1}$ from NMF initialized by random seed $j$;

    generate $S_{kj}$ for $j = 1, 2, \cdots, q$ where $S_{kj}$ is the cosine similarity matrix between topics in $T_j$, $T_{j+1}$;

    **for** *$S_{kj}$, $j = 1, 2, \cdots, q$* **do**

        $LSS_k = \emptyset$;

        add $lss = \min\left(\max(s_{a.}), \max(s_{.b})\right)$ to $LSS_k$, where $s_{ab}$ is the $(a, b)$th entry of the matrix $S_{kj}$;

    **end**

**end**

**return** $k^* = \arg\max_k(median(LSS_k))$;

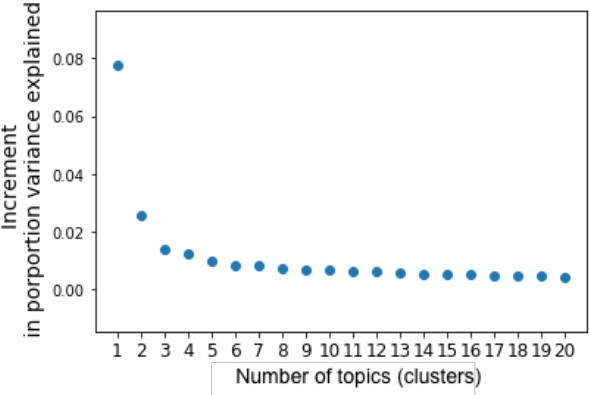

Figure 6: Plot of marginal increment in proportion of variance explained by adding another cluster to split $X$. It is determined that the ideal number of clusters/topics likely lies in the range $[7, 11]$, as this is where the plot starts to level off.

hierarchical structure.

## 5.1 Determining Number of Topics in Each Layer

As previously discussed, the latent topics discovered by NMF are sensitive to the initial state of the algorithm, leading to different dictionaries for each topic. In order to reduce this sensitivity, we seek to find an appropriate number of topics, $k^*$, in each layer such that if a $k^*$-topic NMF is initialized using any two random seeds, the content in the topics discovered should be similar, as measured by cosine similarity. We define this as a *consistent* number of topics. Algorithm 2 summarizes the process to find the "best" number of topics, as defined in this manner, for a corpus matrix $X$.

In Algorithm 2, first the increment in proportion of variance explained by adding one more cluster to split the corpus matrix $X$ is plotted. This is calculated by looking at the singular values of $X$. By examining this plot (Figure 6), a range $[k_1, k_2] = [7, 11]$ in which a potential optimal number of topics, $k^*$ can be found is obtained by noting where the proportion of variance explained starts to level off.

To determine the value of $k^*$ in this range, first, $q + 1$ random seeds are randomly selected, where $q$ is a sufficiently large number. In this case, $q = 30$ was used. For each number of topics $k \in [k_1, k_2]$, topic sets are generated $\{T_j\}_{j=1}^{q+1}$ using each of the $q + 1$ random seeds for initializing NMF.

Then, the cosine similarity is calculated between each of the $k$ topics for every consecutive pair of $T_j$'s. The similarity scores between the topics for each pair $(T_j, T_{j+1})$ are stored in a matrix $S_{kj} \in \mathbb{R}^{k \times k}$. Therefore, $q$ of such matrices are generated for each $k \in [k_1, k_2]$. For a fixed $k$, the minimum of all maximum entries from each column and row of each similarity matrix $S_{kj}$ is defined to be least seed similarity ($lss$) score for that $k$. The set containing the $q$, $lss$ scores for a given number of topics $k$ is denoted $LSS_k$. A consistent number of topics should have an overall high similarity between the topics generated for each seed. Therefore, we choose $k^*$ in $[k_1, k_2]$ to be the "best" number of topics if the median of all its $lss$ scores is the highest.

The boxplot in Figure 7 shows the distribution of the $lss$ scores for $k$ in $[7, 11]$. In this case, 8 is chosen as the "best" number of topics since it results in the highest median $lss$ score.

## 5.2 Implementation of Hierarchical NMF

A hierarchical NMF (see Algorithm 1) is applied to cluster the articles, where the number of topics in each layer is determined by Algorithm 2. The hierarchical tree structure is established from top to bottom and consists of three layers on this data set (see Figure 1).

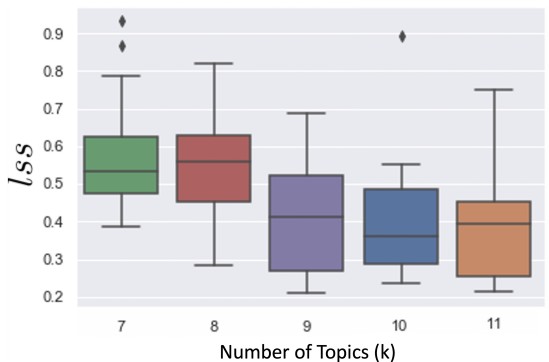

Figure 7: Box plot of $LSS_k$: Topic number 8 is the "best" as it has the highest median $lss$ (least seed similarity) score and should be expected to yield consistent results with random seeds.

To generate topics in the first layer, NMF is applied to the matrix $X$ containing all the vectorized articles, resulting in a factorization with 8 topics, as determined by Algorithm 2. Next, a threshold $\alpha$ (in this case, $\alpha = 0.05$) is chosen, and the articles in $X$ are assigned into a topic class $X_1, \cdots, X_8$ if their corresponding document-topic correlation in the $H$ matrix is greater than $\alpha$. Note that by this definition, one article could be assigned to one or more topic class. After this, any articles not classified to one of the 8 topics are assigned to the "Extra Document" corpus, $X_e$. Now, the second layer of the tree consists of text corpora $X_1, \cdots, X_8$.

For each $X_i$, $i = 1, 2, \cdots, 8$ in the second layer, the topic is further subdivided into a third layer if the number of articles assigned to a topic class $i$ is more than some $m$ (in this analysis, we chose $m = 1400$). If it is determined that text corpus $X_i$ needs to be divided further using NMF, the number of subtopics is chosen by Algorithm 2 and again, articles from $X_i$ are assigned to each subtopic based on the threshold $\alpha$. As before, any articles that do not receive a classification are assigned to $X_e$. This process is continued for each level in the tree until each leaf contains no more than $m$ articles.

Finally, the cosine similarity between each article in $X_e$ and the dictionary associated to each leaf (topic in the lowest layer in a given branch) is calculated. Note that the dictionary of a leaf is a column of the $W$ matrix of its parent topic. Then the articles in $X_e$ are assigned to the leaf with the highest cosine similarity. After this reassignment, the number of articles associated with each leaf is calculated again, and any leaves containing more than $m$ articles are further subdivided.

We note that in this framework, newly published papers could be added to the tree by first assigning them to $X_e$ and then distributing them as described above. However, since the addition of new papers may also necessitate the introduction of new topics, future work includes extending the tool to an online version that would allow for new topics to be added as new papers appear.

## 6   Conclusion and Future Work

HNMF is used to organize existing literature on coronaviruses and pandemics, and early literature on COVID-19 into an interactive structure easily searchable by researchers and available to use through a corresponding website. The topics discovered by HNMF reveal that early research of interest to the COVID-19 research community divides into diverse areas such as research related to other coronaviruses, research related to other respiratory diseases, virology and genetic research, as well as research relating to the public health response. A topic coherence metric reveals that the topics discovered are consistent and semantically meaningful, while a topic similarity metric reveals that the topics differ sufficiently from one another to allow for a diversity of choice and areas of interest on the part of the user.

In the future, we hope to regularly update the hierarchical structure as well as the associated website as new research papers are added, both by adding new papers and by adding and deleting classifications as new research topics emerge. We hope to do this using an online version of the HNMF algorithm such as the one in Tu et al. (2018).

## Acknowledgments

Needell and Rebrova appreciate the support of NSF BIGDATA #1740325 and NSF DMS #2011140.

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

# 7 Appendices

## 7.1 Keywords Removed

Here is the list of top 30 keywords from the topic which is identified as not the content of the publishing information : without, also, biorxiv, perpetuity copyright, ccbyncnd international, ccbyncnd, peerreviewed copyright, perpetuity peerreviewed, medrxiv preprint, made available, preprint peerreviewed, available authorfunder, license made, international license, granted,perpetuity, display preprint, preprint perpetuity, license display, authorfunder granted, granted medrxiv, medrxivlicense, peerreviewed, holder, authorfunder, copyright, copyright holder, holder preprint, covid[3], license, medrxiv, preprint.

## 7.2 Topic Keywords

Following tables list ten most probable keywords associated with each topic and subtopic in the tree generated by HNMF and the $C_V$ coherence score associated with each. These keywords are visible to website users to enable them to make choices to navigate through the tree. Note that for the first layer we gave suggested topic titles. Not being experts in the field, these are only suggestions to give an idea of the types of research someone may be looking for within that topic.

---

[3]We note that we performed a semantic comparison of topics generated using our algorithm both including and removing the word "covid". No significant differences in topic interpretation were found between the two results indicating that including the word did not add additional information to the topic modeling in this case.

| Topic Number | Key Words | Possible Topic |
|---|---|---|
| 1 | merscov, bat, sarscov, camel, mers, merscov infection, ace, human, virus, rbd | **Coronaviruses affecting Humans** |
| 2 | cell, mouse, expression, infection, virus, gene, response, viral, cytokine, immune | **Cellular immune response to viral infection** |
| 3 | sequence, gene, genome, strain, rna, primer, ibv, nucleotide, sample, using | **Genetic characteristics of the virus** |
| 4 | protein, binding, residue, peptide, domain, structure, compound, membrane, cell, activity | **Cell-protein interaction** |
| 5 | virus, influenza, child, rsv, respiratory, infection, viral, sample, assay, detection | **Detection and biological study of respiratory viruses** |
| 6 | patient, hospital, study, pneumonia, clinical, day, infection, treatment, case, symptom | **Clinical and hospital studies (esp. of respiratory illnesses)** |
| 7 | health, model, disease, case, epidemic, outbreak, public, country, population, transmission | **Infection models and experiments related to public health** |
| 8 | vaccine, antibody, pedv, mouse, serum, antigen, pig, strain, protein, response | **Vaccine development (esp. of the coronavirus PEDV)** |

Table 2: The top 10 keywords associated with each of the 8 topics in the first layer of the tree

| Topic Number | Key Words | $C_V$ |
|---|---|---|
| 1-1 | camel, merscov, dromedary, human, sample, dromedary camel, animal, herd, sequence, study | 0.56 |
| 1-2 | sarscov, ace, protein, ncov, rbd, binding, sars, residue, sequence, virus | 0.61 |
| 1-3 | patient, case, merscov, infection, mers, hospital, outbreak, disease, respiratory, day | 0.63 |
| 1-4 | bat, specie, virus, sequence, bat specie, human, sample, host, covs, study | 0.57 |
| 1-5 | cell, merscov, mouse, protein, antibody, vaccine, virus, response, infection, serum | 0.55 |

Table 3: The top 10 keywords associated with each of the subtopics of Topic 1 in the $2^{nd}$ layer of the tree and the $C_V$ coherence score

| Topic Number | Key Words | $C_V$ |
|---|---|---|
| 2-1 | infection, response, lung, immune, cytokine, mouse, tlr, ifn, virus, macrophage | 0.59 |
| 2-2 | virus, cell, viral, infection, infected, replication, vero, culture, antiviral, vero cell | 0.61 |
| 2-3 | cell, antibody, tumor, antigen, culture, human, line, cell line, surface, patient | 0.41 |
| 2-4 | protein, expression, gene, cell, figure, pathway, using, sirna, activity, level | 0.55 |
| 2-5 | mouse, cns, brain, day, demyelination, cell, astrocyte, mhv, day pi, spinal | 0.74 |

Table 4: The top 10 keywords associated with each of the subtopics of Topic 2 in the $2^{nd}$ layer of the tree and the $C_V$ coherence score

| Topic Number | Key Words | $C_V$ |
|---|---|---|
| 3-1 | bat, specie, covs, cov, bat specie, virus, sequence, human, sample, coronaviruses | 0.59 |
| 3-2 | gene, rna, protein, cell, expression, mrna, sequence, codon, virus, orf | 0.61 |
| 3-3 | sequence, virus, genome, read, viral, analysis, human, tree, using, specie | 0.47 |
| 3-4 | sample, primer, assay, pcr, probe, detection, dna, virus, reaction, amplification | 0.70 |
| 3-5 | strain, pedv, sequence, pedv strain, aa, vp, nt, diarrhea, gene, china | 0.55 |
| 3-6 | ibv, strain, chicken, vaccine, ibv strain, isolates, virus, bird, gene, flock | 0.72 |

Table 5: The top 10 keywords associated with each of the subtopics of Topic 3 in the $2^{nd}$ layer of the tree and the $C_V$ coherence score

| Topic Number | Key Words | $C_V$ |
|---|---|---|
| 4-1 | compound, activity, inhibitor, drug, derivative, mmol, docking, pro, protease, ic | 0.62 |
| 4-2 | nsp, rna, sarscov, nsp nsp, mm, replication, protein, activity, domain, rdrp | 0.61 |
| 4-3 | protein, sequence, interaction, gene, analysis, also, study, function, method, used | 0.40 |
| 4-4 | protein, cell, antibody, mm, sarscov, using, min, expression, serum, recombinant | 0.61 |
| 4-5 | virus, cell, viral, rna, replication, infection, membrane, host, hcv, er | 0.64 |
| 4-6 | peptide, residue, structure, sarscov, binding, fusion, domain, figure, sequence, hr | 0.63 |

Table 6: The top 10 keywords associated with each of the subtopics of Topic 4 in the $2^{nd}$ layer of the tree and the $C_V$ coherence score

| Topic Number | Key Words | $C_V$ |
|---|---|---|
| 5-1 | assay, pcr, sample, detection, primer, sensitivity, specimen, method, amplification, probe | 0.74 |
| 5-2 | child, rsv, hmpv, infection, study, hbov, asthma, infant, respiratory, age | 0.79 |
| 5-3 | influenza, ili, virus, surveillance, sari, case, influenza virus, year, study, season | 0.66 |
| 5-4 | patient, respiratory, infection, study, pneumonia, viral, virus, bacterial, pneumoniae, pathogen | 0.60 |
| 5-5 | virus, cell, influenza, infection, influenza virus, protein, viral, antibody, ha, mouse | 0.56 |
| 5-6 | virus, sample, sequence, human, read, hbov, genome, viral, sequencing, study | 0.52 |

Table 7: The top 10 keywords associated with each of the subtopics of Topic 5 in the $2^{nd}$ layer of the tree and the $C_V$ coherence score

| Topic Number | Key Words | $C_V$ |
|---|---|---|
| 6-1 | patient, sarscov, display, ct, case, reserved reuse, allowed permission, reuse allowed, permission display, wuhan | 0.61 |
| 6-2 | cap, patient, antibiotic, pneumonia, study, pneumoniae, bacterial, pathogen, infection, culture | 0.64 |
| 6-3 | virus, infection, respiratory, patient, viral, child, rsv, influenza, study, respiratory virus | 0.59 |
| 6-4 | sars, patient, hospital, contact, case, transmission, sars patient, outbreak, staff, care | 0.67 |
| 6-5 | patient, study, treatment, cell, disease, group, level, lung, therapy, day | 0.43 |

Table 8: The top 10 keywords associated with each of the subtopics of Topic 6 in the $2^{nd}$ layer of the tree and the $C_V$ coherence score

| Topic Number | Key Words | $C_V$ |
|---|---|---|
| 7-1 | health, public, public health, care, patient, disease, emergency, hospital, system, response | 0.67 |
| 7-2 | disease, animal, human, virus, pathogen, specie, host, infection, vaccine, zoonotic | 0.61 |
| 7-3 | model, individual, epidemic, network, parameter, infected, node, contact, number, rate | 0.57 |
| 7-4 | data, model, study, used, analysis, case, variable, using, method, time | 0.44 |
| 7-5 | case, available display, international made, display, made available, day, wuhan, number, china, international | 0.60 |

Table 9: The top 10 keywords associated with each of the subtopics of Topic 7 in the $2^{nd}$ layer of the tree and the $C_V$ coherence score

| Topic Number | Key Words | $C_V$ |
|---|---|---|
| 8-1 | pig, serum, antibody, virus, piglet, group, sample, day, prrsv, tgev | 0.58 |
| 8-2 | mouse, cell, response, group, merscov, immunized, immunization, dna, protein, antibody | 0.55 |
| 8-3 | vaccine, virus, response, influenza, disease, vaccination, immune, human, development, antigen | 0.51 |
| 8-4 | pedv, strain, piglet, pedv strain, ped, cell, gene, diarrhea, sequence, pig | 0.56 |
| 8-5 | protein, antibody, epitope, mabs, peptide, serum, sarscov, mab, elisa, binding | 0.61 |
| 8-6 | ibv, chicken, strain, bird, ibv strain, group, virus, vaccine, ib, egg | 0.72 |

Table 10: The top 10 keywords associated with each of the subtopics of Topic 8 in the $2^{nd}$ layer of the tree and the $C_V$ coherence score

| Topic Number | Key Words | C_V |
|---|---|---|
| 7-1-1 | china, research, chinese, disaster, government, social, also, development, policy, people | 0.62 |
| 7-1-2 | patient, hospital, care, infection, staff, medical, health care, nurse, physician, healthcare | 0.66 |
| 7-1-3 | health, public health, public, disease, surveillance, country, system, global, laboratory, outbreak | 0.65 |
| 7-1-4 | pandemic, influenza, participant, respondent, sars, study, outbreak, risk, public, information | 0.55 |

Table 11: The top 10 keywords associated with each of the subtopics of Topic 7-1 in the $3^{rd}$ layer of the tree and the $C_V$ coherence score

| Topic Number | Key Words | C_V |
|---|---|---|
| 7-4-1 | study, risk, participant, age, influenza, respondent, factor, country, health, population | 0.52 |
| 7-4-2 | sample, rat, cell, group, animal, cat, used, using, study, protein | 0.35 |
| 7-4-3 | model, data, case, outbreak, surveillance, disease, epidemic, transmission, influenza, time | 0.54 |
| 7-4-4 | air, particle, concentration, wind, velocity, ventilation, flow, airflow, temperature, room | 0.73 |
| 7-4-5 | calf, diarrhea, farm, colostrum, milk, fecal, cow, dairy, herd, day | 0.72 |

Table 12: The top 10 keywords associated with each of the subtopics of Topic 7-4 in the $3^{rd}$ layer of the tree and the $C_V$ coherence score