# OpenReview forum: "COVID-19 Literature Topic-Based Search via Hierarchical NMF"
_EMNLP/2020/Workshop/NLP-COVID — NLP-COVID19-EMNLP Oral_

### Official Review · AnonReviewer1 · 2020-09-23
**Nice paper with practical/useful results**

**Rating:** 7
**Confidence:** 4

**Review:**

The paper describes the creation of a dataset with scientific papers about COVID-19 that have been divided in a three-layer structure based on the topics covered. The authors have used hierarchical nonnegative matrix factorization to organize the topics. A total of 52 major topics were found. The authors evaluated the results trough different measurements. The work is of high interest given the ability of "classify" papers based on specific and concrete topics, allowing to filter the large amount of literature that has been produced in the last months with respect to COVID-19, being an useful tool. My only concern is that I cannot see where is possible to download the dataset used to create the topic structure, or if could be possible to use the tool for new articles classification that will appear in the future.

---

> ### Author Response · Authors · 2020-09-28
> **Response to the two concerns raised by the review**
>
> 1. We thank the reviewer for pointing out the importance of having the whole dataset available. We have provided a link on our website http://covid-19-literature-clustering.net/about/ where readers can download the corpus we used to construct the hierarchical tree structure.
> 2. The question of whether the tool could be used to classify future articles is a good one. In the context of our current framework, if a new paper becomes available we could assign the new paper into the current tree structure in the same way that articles in the "extra article" set, $X_e$, are distributed (see Section 5.2). However, since the addition of new papers may also  necessitate the introduction of new topics, our future work, as mentioned in the paper, includes extending the tool to an online version (see, e.g., Tu  et  al.  (2018)) that would allow for new topics to be added as new papers appear.

---

### Official Review · AnonReviewer3 · 2020-09-25
**Useful and well-written paper**

**Rating:** 7
**Confidence:** 4

**Review:**

This paper presents a topic modelling system based on hierarchical nonnegative matrix factorisation. The paper itself is very well written and structured, there is a clear logical progression between sections and ideas. The accompanying website is easy to use, though can be improved in terms of visual representation.
Though the paper does not introduce a novel method, it makes several meaningful contributions including the following:
1. The authors augment Kaggle's COVID-19 dataset by additionally searching 4 databases, removing non-English publications and articles without abstract or full text, which helps to build more meaningful and relevant topic models
2. The resulting model helps to avoid flat structure and discover meaningful subtopics on as many as three levels of nesting. Some subtopics can belong to several major topics and the authors provide examples to justify this. Overall, such hierarchical structure can be useful for researchers to explore the topics of interest in more detail.
3. The topics are thoroughly analysed in terms of major issues discussed in COVID-19 literature, thus presenting a valid use-case scenario.

However, there are some weak points in terms of reproducibility and implementation:

1. The corpus is not provided, and it is impossible to reproduce based on the information in the article.
2. It is not clear how the texts were preprocessed. The authors state that "words deem to be irrelevant such as “copyright” or “et al” are removes”, but they provide no criteria for this judgment, such as a publicly available stop-words list or, say, 50 most frequent words in the data set. Again, this makes the paper impossible to reproduce and can potentially affect the results of the topic model. For example, if the authors manually checked the texts for “irrelevant” words and then removed them, that would lead to much more relevant and coherent results and thus we’d overestimate what the model can do.
3. More of a suggestion - though the topics look coherent by “eyeballing”, you might consider using a more reliable coherence metric than Mimno et al. 2011, which has the worst correlation with human evaluation (Röder, M., Both, A., & Hinneburg, A. (2015, February). Exploring the space of topic coherence measures. In Proceedings of the eighth ACM international conference on Web search and data mining (pp. 399-408).)

---

> ### Author Response · Authors · 2020-09-28
> **Response to the comments on the weak points in terms of reproducibility and implementation**
>
> We thank the reviewer for their very helpful feedback and comments. We respond to the three concerns listed in the review in the same order the reviewer listed them:
> 1. We absolutely agree that it is important for our work to be reproducible and thank the reviewer for pointing out that the work was not reproducible as written. As such, we have provided a link on out website http://covid-19-literature-clustering.net/about/ where readers can download the corpus after the preprocessing that we used to construct hierarchical tree structure.
> 2. We thank the reviewer for this helpful comment. Should our paper be accepted, we would gladly further expand on how the preprocessing was done for the final version, to guarantee the reproducibility of our work.
> We will briefly summarize the process here.The first criteria for removing words in our initial pre-processing was the publicly available and commonly used NLTK English stopwords list (Birdet al., 2009). After removal of these stopwords, we noticed that the topics generated using our algorithm frequently included words that, upon further investigation, were found to be  part of the copyright or publishing information and not the content of the articles themselves and so we removed them as well. We will include the complete list of removed words and an explanation in the final version of the paper, should it be accepted.
> 3. We thank the reviewer for this helpful suggestion. We have read the paper suggested (Röder, Both, \& Hinneburg, 2015) and are working on implementing the $C_V$ coherence metric, the one most highly recommended by the paper. We hope to add to or replace the coherence results from the Mimno metric with these results if our paper should be accepted.

---

### Official Review · AnonReviewer4 · 2020-09-30
**Conventional topic inference work with delicate algorithmic design form a nice pipeline to trace Covid-19 literature**

**Rating:** 7
**Confidence:** 3

**Review:**

We might say, neither NMF nor HNMF is a novel idea in topic inference, regardless the known LDA in history and powerful Transformers variances (If one use BERT or SciBERT to augment semantics in the research) in current days. However, this work is fairly well designed, and I believe it reaches sufficient quality level published in EMNLP workshop.

Strong points:
1. The overall scientific design is reasonable. The manuscript is fairly well-written.
2. In the implementation section, algorithm 2 is carefully designed and it assures the reasonability of topic amount selection.

Weak points:
1. It seems very doable for this work to provide "raw data/codes" reproducibility, but authors did not mention it. ---This is just a humble suggestion, though.
2. Introducing Word2vec in WMD makes sense, that is true. However, in the meantime, currently there are quite a lot new focuses in semantics representation of phrase or sentence, instead of single lexicon. Conventional method could be augmented with new idea so as to explore more in depth.
3. Somehow, some baseline methods could be introduced, implemented and tested. Before jumping into the analysis of the final 52 (sub-)topics, it is of utmost importance to make the HNMF algorithmic part convincing.

---

> ### Author Response · Authors · 2020-10-10
> **Response to Concerns Raised by the Reviewer**
>
> We thank the reviewer for their very helpful feedback and comments. We respond to the three concerns listed in the review in the same order the reviewer listed them:
> 1. We thank the reviewer for this suggestion and agree that the reproducibility of the work if very important. We have provided a link on our website to the original (cleaned) dataset (http://covid-19-literature-clustering.net/about). We also plan to provide a GitHub link to our code on our website in the future.
> 2. We thank the reviewer for this helpful note. We acknowledge that there are indeed some state of the art language models such as BERT and ELMO that take into account the contextual meaning instead of a single lexicon as Word2Vec.
> However, we note that the topics extracted from the NMF is a bag of words with weights that do not necessarily have contextual meaning. For this reason, we chose to use the Word2Vec with WMD for our application. We have made a note of this in the final version of our manuscript.
> 3. We thank the reviewer for this helpful suggestion. In the final version of our paper, we have added some further justification in Section 3.2 as to why the HNMF algorithm is more advantageous than traditional NMF for this application. After careful consideration and due to length restrictions, we chose not to include baseline examples of the HNMF algorithm. We decided that the application of the algorithm and associated results are more pertinent to the goals of the workshop and so chose to focus our attention there. However, we did add a reference to a paper that describes an online version of HNMF that is very similar to our own and provides several baseline examples to demonstrate the effectiveness of HNMF in topic detection.